# Different Temperature Treatments of Millet Grains Affect the Biological Activity of Protein Hydrolyzates and Peptide Fractions

**DOI:** 10.3390/nu11030550

**Published:** 2019-03-05

**Authors:** Monika Karaś, Anna Jakubczyk, Urszula Szymanowska, Krystyna Jęderka, Sławomir Lewicki, Urszula Złotek

**Affiliations:** 1Department of Biochemistry and Food Chemistry, University of Life Sciences, Skromna 8, 20-704 Lublin, Poland; monika.karas@up.lublin.pl (M.K.); urszula.szymanowska@up.lublin.pl (U.S.); urszula.zlotek@up.lublin.pl (U.Z.); 2Department of Regenerative Medicine and Cell Biology, Military Institute of Hygiene and Epidemiology, Kozielska 4, 01-163 Warsaw, Poland; krystyna.jederka@wihe.pl (K.J.); lewickis@gmail.com (S.L.)

**Keywords:** peptides, metabolic syndrome, millet, endothelial cells, HECa10 line

## Abstract

The objective of this study was to analyze millet protein hydrolyzates and peptide fractions with molecular mass under 3.0 kDa obtained from grains treated with different temperature values as inhibitors of angiotensin-converting enzyme (ACE), α-amylase, and α-glucosidase activity. The protein fractions were hydrolyzed in vitro in gastrointestinal conditions and the highest degree of hydrolysis was noted for globulin 7S obtained from control grains (98.33%). All samples were characterized by a high peptide bioaccessibility index, which was 23.89 for peptides obtained from globulin 11S after treatment with 100 °C. The highest peptide bioavailability index was noted for peptides obtained from globulin 11S after the treatment with 65 °C (2.12). The highest potential metabolic syndrome inhibitory effect was determined for peptide fractions obtained from the prolamin control (IC_50_ for ACE and α-amylase was 0.42 and 0.11 mg/mL, respectively) and after the 100 °C treatment (IC_50_ for ACE and α-glucosidase was 0.33 and 0.12 mg/mL, respectively) and from globulin 11S after the 65 °C treatment (IC_50_ 0.38 and 0.05 for ACE and α-glucosidase, respectively). The effect of these samples on endothelial cell HECa10 was determined. The sequences of potential inhibitory peptides were identified as GEHGGAGMGGGQFQPV, EQGFLPGPEESGR, RLARAGLAQ, YGNPVGGVGH, and GNPVGGVGHGTTGT.

## 1. Introduction

Millet is grown all over the world due to the low cost of cultivation, its biodiversity—pearl millet (*Pennisetum glaucum*), foxtail millet (*Setaria italica*), proso millet (*Panicum miliaceum*), and finger millet (*Eleusine coracana*)—and high biological properties. Probably, it is the first cereal cultivated by man and the first reports about the cultivation of millet date back to about 5550 BC [1]. It should be noted that millet does not contain gluten, which cannot be consumed by people with celiac disease (life-long intolerance to ingested gluten) as it causes autoimmune disorders affecting the gastrointestinal system [2]. Millet protein is rich in essential amino acids except tryptophan and lysine, which are generally limiting amino acids in cereals and legumes. On the other hand, the proteins are relatively rich in sulfur-containing amino acids such as cysteine and methionine [3]. Millet grains are characterized by low fat content (1.5–5%), but they are rich in carbohydrates (60–70%) and contain 7–12% of proteins and 2–7% of fiber. They are a good source of vitamins (especially vitamin B: thiamine, folacin, niacin, and riboflavin) and minerals such as magnesium, iron, and calcium. Moreover, they contain some essential fatty acids like linoleic, oleic, and palmitic acids found in a free form [4].

Millets can be used in the management of type 2 diabetes due to their hypoglycemic property; their antioxidant activities have been reported as well [5]. Moreover, according to Chen et al. [6], foxtail millet hydrolyzates can decrease hypertension and the risk of cardiovascular diseases, which are part of metabolic syndrome (MS). It is defined as a multiplex risk factor for atherosclerotic cardiovascular disease and type 2 diabetes and consists of five main factors, such as atherogenic dyslipidemia, high blood pressure, dysglycemia, a pro-thrombotic state, and a pro-inflammatory state [7]. The pathogenesis of MS is still not clear; however, it is known to be related to metabolic disorders leading to the development of obesity, insulin resistance, and hypertension. One of the causes of the occurrence of diabetes mellitus is hyperglycemia, which is related to insufficient or inefficient insulin secretion and alterations in carbohydrate, protein, and lipid metabolism. One of the methods for preventing postprandial hyperglycemia is inhibition of carbohydrate absorption after meals. In the first step, polysaccharides in the gastrointestinal tract are hydrolyzed by α-amylase to dextrins or oligosaccharides, which are a substrate for the action of α-glucosidase, which leads to the release of a large amount of glucose. Therefore, inhibition of enzymes involved in polysaccharide degradation can reduce postprandial hyperglycemia [8].

Long-lasting high blood pressure can lead to the development of hypertension and cardiovascular diseases, such as stroke, coronary heart disease, and peripheral arterial disease. The main role in maintaining normal blood pressure is played by the angiotensin-converting enzyme (ACE). Its excessive activity causes vascular muscle hyperplasia, reduction of vascular lumen, and development of hypertension [9].

Nowadays, some synthetic inhibitors of enzymes causing hydrolysis of polysaccharides (e.g., acarbose) and inhibition of ACE activity (e.g., captopril or enalapril) have been widely used for pharmacological treatment of hyperglycemia, hypertension, and heart failure but they may cause side effects such as liver disorders, flatulence and abdominal cramping, rash, and cough [10]. It is known that bioactive compounds, including peptides derived from food, prevent the development of cardiovascular diseases by modulation of the angiogenesis process, given the fact that endothelial dysfunction plays a key role in the initiation and pathogenesis of atherosclerosis [11]. Angiogenesis is involved in endothelial cell attachment based on membrane degradation, synthesis, and migration and proliferation of cells. It participates in tissue formation and wound healing as well as the pathogenesis of cancers, systemic lupus erythematosus, Takayasu’s arteritis, or other autoimmune diseases. Therefore, identification of conditions and factors influencing blood vessel formation is one of the major issues in prevention of diseases [12]. Several studies have demonstrated the beneficial effects of peptides derived from food on cardiovascular disorders, i.e., improvement of endothelial activity [13].

The aim of this study was to investigate the protein fraction obtained from millet grains after cooking at different temperature values (65 °C and 100 °C) as a source of protein hydrolyzates with potential antidiabetic and antihypertension effects. Moreover, the influence of the protein hydrolyzates and peptide fractions on the viability of endothelial cells from the HECa10 line was determined.

## 2. Materials and Methods

### 2.1. Materials

The millet grains (*Panicum miliaceum* L.) were purchased from the Horticulture and Nursery Industry (PNOS) in Ożarów Mazowiecki, Poland. *Panicum miliaceum* L. is one of the oldest cultivated and first domesticated crops.

### 2.2. Millet Grain Heating

The millet grains were added to distilled water at a grain/water ratio 1:2 (*w*/*v*). The samples were heated at two temperature values, i.e., 65 °C and 100 °C, for 30 and 15 min, respectively. Grains that were not exposed to the heating process were used as a control sample. The grains were lyophilized. All types of grains were obtained in triplicate.

### 2.3. Protein Fractionation

Heated grains were ground with a laboratory mill and the flour was defatted with hexane at a flour/hexane ratio of 1:10 (*w*/*v*). The samples were stirred for 1 h at 4 °C and next centrifuged at 8000× *g* for 20 min. The supernatants were dried in a laboratory dryer at 25 °C. Defatted dry flours were kept at 4 °C until use. The millet seed protein extraction was carried out according to Silva-Sánchez et al. [14]. All fractions were obtained in triplicate.

### 2.4. In Vitro Hydrolysis of Proteins and Preparation of the Peptide Fraction

All protein fractions were hydrolyzed in vitro in gastrointestinal conditions according to the method described previously [15]. Peptide fractions <3.0 kDa were obtained with Amicon Ultra-15 Centrifugal Filter Units, Merck Millipore (Membrane NMWL, 3 kDa).

### 2.5. Degree of Hydrolysis (DH)

In each of the hydrolysis steps, the degree of hydrolysis was determined with the trinitrobenzenesulfonic acid (TNBS) method using L-leucine as a standard [16].

### 2.6. Potential Bioaccessibility and Bioavailability of Peptides Obtained from Millet Proteins

Theoretical calculation of the nutritional potential was based on the index described by Gawlik-Dziki et al. [17]. The peptide bioaccessibility index (PAC), which is an indicator of the bioaccessibility of peptides, was expressed as:PAC = Cph/CpbCph–peptide content in the hydrolyzateCpb–peptide content in the sample before hydrolysis

The peptide bioavailability index (PAV), which is an indicator of the bioavailability of peptides, was expressed as:PAV = Cpa/CphCpa–peptide content after the absorption processCph–peptide content in the hydrolyzate

### 2.7. Enzyme Inhibitory Activity Assay

#### 2.7.1. Angiotensin-Converting Enzyme (ACE) Inhibitory Assay

The ACE inhibitory activity of the hydrolyzates and peptide fractions was measured with the spectrophotometric method using BioTek Microplate Readers. For the ACE activity assay, 5 μL of an ACE solution was added to 5 μL of borate buffer pH = 8.3 with 0.3 M NaCl. After adding 5 μL of 5 mM HHL, the reaction was carried out for 1 h at 37 °C. The reaction was stopped by adding 70 μL of 0.1 M borate buffer pH = 8.3 with 0.2 M NaOH. Next, 150 μL of a 1 mM o-phthalaldehyde (OPA) solution was added. The absorbance at 390 nm was measured. The inhibitory activity assays were performed in 5 μL of samples with the same reaction conditions as those described above.

The ACE inhibition was determined as follows:ACE inhibition (%) = [1 − ((A1 − A2)/A3)] × 100, where:A1 is the absorbance of the sample with ACE and the inhibitor,A2 is the absorbance of the sample with inhibitor without ACE,A3 is the absorbance of the sample with ACE and without the inhibitor.

#### 2.7.2. α-Amylase Inhibitory Assay

α-Amylase inhibitory activity (αAI) of the protein hydrolyzates and peptide fractions was measured according to the method described by Świeca et al. [18]. α-Amylase from hog pancreas (50 U/mg) was dissolved in the 100 mM phosphate buffer (containing 6 mM NaCl, pH 7.0). To measure the α-amylase inhibitory activity, a mixture of 25 μL of α-amylase solution and 25 μL of sample was first incubated at 40 °C for 5 min. Then, 50 μL of 1% (*w*/*v*) soluble starch (dissolved in 100 mM phosphate buffer containing 6 mM NaCl, pH 7) was added. After 10 min, the reaction was stopped by adding 100 μL of 3,5-dinitrosalicylic acid (DNS) and was heated for 10 min. The mixture was then made up to 300 μL with double distilled water. After that the absorbance at 540 nm was measured using BioTek Microplate Readers. The final results were compared with the activity of the same amount of enzyme without the inhibitor. All assays were carried out in triplicate. For IC_50_ value determination, the inhibitory activity for four concentrations of samples was investigated.

#### 2.7.3. α-Glucosidase Inhibitory Assay (αGIA)

The αGIA was measured with the method described by Jakubczyk et al. [19]. Firstly, 10 μL of α-glucosidase (1 U/mL) and 20 μL 35 mmol/L p-nitrophenol were added to 0.5 mL of 0.1 mol/L phosphor buffer pH = 6.8. The reaction was incubated at 37 °C for 20 min. The α-glucosidase activity was observed as an increase in absorbance at 405 nm. For the αGIA measurement, 10 μL of α-glucosidase (1 U/mL) and 50 μL of the sample were added to 0.45 mL of 0.1 mol/L phosphor buffer pH = 6.8. After the incubation reaction at 37 °C for 5 min, 20 μL of 35 mmol/L p-nitrophenol was added. The reaction was incubated at 37 °C for 20 min. The α-glucosidase activity in the sample was expressed as an increase in absorbance at 405 nm.

The IC_50_ value defined as the concentration of the extract inhibiting 50% of the enzyme activity was determined by measuring the enzyme inhibitory activity and peptide contents in each sample. The IC_50_ value was calculated from the plotted graph of the inhibition activity for the five different peptide concentrations.

### 2.8. Effect of Protein Hydrolyzates and Peptide Fractions on the Metabolism of Endothelial Cells (HECa10 Line)

The studies consisted of an assessment of the impact of the protein hydrolyzates and peptide fractions isolated from millet on the cell count (MTT and neutral red (NR) assay), viability and type of cell death, and cell cycle of endothelial cells (HECa10 line). We used two methods (MTT and NR tests) that measure different metabolic markers to estimate the number of viable cells in the culture to obtain more adequate results (each cell viability assay has its own set of advantages and disadvantages) [20].

#### 2.8.1. MTT Test

The assay was performed as previously described [21]. Briefly, the cells were seeded in a 96-well culture plate at a concentration of 1 × 10^4^ cells/well. Twenty-four hours after seeding, the cells were rinsed twice with phosphate-buffered saline (PBS) (Life Technologies, Warsaw, Poland) and resuspended in fresh growth medium. Peptide fractions were added at concentrations of 0 (control), 0.1, 1, 5, 10, 50, and 100 µg/mL. After 24 h incubation with the proteins, assessment of cell metabolic activity based on MTT tests was performed. The absorbance was measured at 570 nm with a FLUOstar Omega reader (BMG Labtech, Ortenberg, Germany). The results are presented as the percentage of the control values (mean ± SD). Each assay was performed in triplicate (*n* = 18).

#### 2.8.2. NR Test

The assay was performed as previously described [22]. Briefly, the cells were seeded in 96-well culture plate at a concentration of 1 × 10^4^ cells/well. Twenty-four hours after seeding, the cells were rinsed twice with PBS (Life Technologies, Warsaw, Poland) and resuspended in fresh growth medium. Peptide fractions were added at concentrations of 0 (control), 0.1, 1, 5, 10, 50, and 100 µg/mL. After 24 h incubation with the proteins, the uptake of the neutral red (NR) was assessed. The absorption was measured at a wavelength of 540 nm with the background cutoff at 690 nm (FLUOstar Omega, BMG Labtech, Ortenberg, Germany). The results are presented as the percentage of the control values (mean ±SD). Each assay was performed in triplicate (*n* = 18).

#### 2.8.3. Cell Viability and Type of Cell Death

The assay was performed as previously described by Leśniak et al. [22]. In brief, the cells were seeded at a density of 7 × 10^5^ in a 6-well plate. Twenty-four hours later, the cells (approx. 60% confluence) were treated with different peptide fractions at concentrations of 0 (control), 0.1, 1, 5, 10, 50, and 100 µg/mL for 24 h. After the treatment, floating cells were collected for further analysis and adherent cells were trypsinized. Subsequently, all cells (floating and adherent) were washed twice with PBS, centrifuged (500× *g*, 5 min), and resuspended in binding buffer with an annexin V-FITC antibody (5 µL, eBioscience, Warsaw, Poland) and propidium iodide (PI, 5 µL, Sigma Aldrich, Poznan, Poland). After 20 min incubation, the cells were washed twice with PBS and analyzed with flow cytometry (FACS Calibur, BD, San Jose, CA, USA). Evaluation of cell apoptosis, necrosis, and viability was performed using Cell Quest software (BD, San Jose, CA, USA). The assay was performed in triplicate (*n* = 6).

#### 2.8.4. Cell Cycle

The cells were seeded at a density of 7 × 10^5^ in a 6-well plate and cultured to obtain 60–70% confluence. Then the cells were treated with different peptide fractions: 0 (control), 0.1, 1, 5, 10, 50, and 100 µg/mL for 24 h. After the treatment, floating cells were collected for further analysis and adherent cells were trypsinized. Floating and adherent cells were combined, washed twice with PBS, centrifuged (500× *g*/5 min), and fixed with 70% cold methanol at −20 °C for 24 h. Then, methanol was removed (centrifugation, 500× *g*/5 min) and the cell pellets were resuspended in PBS with propidium iodide and RNase A solution and incubated for 30 min in the dark at room temperature. The stage of the cell cycle was assessed with flow cytometry (FACS Calibur, BD, San Jose, CA, USA) and calculated using ModFit LT 4.1 software (Verity Software House, Topsham, ME, USA). The assay was performed in triplicate (*n* = 6).

### 2.9. Peptide Separation by Gel Filtration Chromatography

Peptide fractions with the highest inhibitory properties were separated by gel filtration chromatography on Sephadex G10 (column: 1.5 × 30 cm; eluent: distilled water; flow rate: 0.8 mL/min). The absorbance of one-milliliter fractions was monitored at 220 nm and next pooled to determine the inhibitory activity of the enzymes. Fractions with the highest activity were lyophilized and used for further LC-MS-MS/MS analysis.

### 2.10. Identification of Peptides

The peptide fraction was analyzed with LC-MS-MS/MS (liquid chromatography coupled to tandem mass spectrometry) using a Nano-Acquity (Waters) LC system and an Orbitrap Velos mass spectrometer (Thermo Electron Corp., San Jose, CA, USA) according to the method described in a previous study [23]. The analysis was performed by the Mass Spectrometry Laboratory in Warsaw, Poland. The equipment used was sponsored in part by the Centre for Preclinical Research and Technology (CePT), a project co-sponsored by the European Regional Development Fund and Innovative Economy, the National Cohesion Strategy of Poland.

### 2.11. Statistical Analysis

All determinations were performed in triplicate. Statistical analysis was performed using STATISTICA 7.0 software for mean comparison using ANOVA with post-hoc Tukey’s HSD (honestly significant difference) test at the significance level α = 0.05.

Data obtained from the experiment on cells was checked for normality of the distribution (Shapiro–Wilk test). The level of statistical significance was calculated: in the case of normal distribution-one-way ANOVA with Bonferroni correction and student’s *t* test; in another case-non-parametric one-way ANOVA with correction Kruskal–Wallis and Mann-Whitney test. The data was analyzed using the GraphPad Prism program (version 5, GraphPad Software, Inc., La Jolla, CA, USA) at a significance level *p* < 0.05.

## 3. Results

### 3.1. Degree of Hydrolysis and Potential Bioaccessibility (PAC) and Bioavailability (PAV) of Peptides

Proteins obtained from millet seeds subjected to different temperature treatments were digested in vitro in gastrointestinal conditions and DH parameters were determined for these hydrolyzates, as shown in Table 1. The results indicated that the temperature applied during millet seed preparation had an influence on the DH values. It should be noted that, only in the case of the albumin hydrolyzate obtained after the last step of hydrolysis, there were no significant differences between DH determined for the control sample and the samples from the two temperature treatments. Moreover, the highest degree of hydrolysis among all hydrolyzates was noted for the albumin 7S hydrolyzate of the control sample obtained after the use of pancreatin (98.33%). The lowest level (34.47%) of hydrolysis was noted in the case of globulin 11S from millet after the 65 °C treatment. Besides the DH of proteins, there are also other important parameters such as potential bioaccessibility (PAC) and bioavailability (PAV) of peptides, as shown in Table 1. The peptide bioaccessibility index for all samples, which was significantly higher than 1, indicates that the peptides from the millet proteins were highly bioaccessible in vitro. The PAC values were lower than those in the control samples only for the albumin samples, whereas the application of temperature in the other cases contributed to the increase in this factor. The highest bioaccessibility potential was noted for the peptides obtained from globulin 11 after the 100 °C treatment. Although the PAC values were high, the potential peptide bioavailability (PAV) factor confirmed poor bioavailability in vitro in almost all cases. The PAV index was significantly higher than 1 (2.12) only for the peptides obtained from globulin 11S derived from millet after the 65 °C treatment. It should be noted that the use of temperature resulted in an increase in the potential bioavailability of peptides in all cases.

### 3.2. Inhibition of Metabolic Syndrome Enzymes

The hydrolyzates were tested as potential inhibitors of enzymes involved in metabolic syndrome pathogenesis. This activity was determined based on the ACE, α-amylase, and α-glucosidase inhibitory activity expressed as an IC_50_ value, as shown in Table 2. Among the potential ACE inhibitory hydrolyzates, the lowest IC_50_ values were noted for samples obtained after the temperature treatment. The significantly highest ACE inhibitory activity was noted for hydrolyzates obtained from millet globulin 11S after the 65 °C treatment (IC_50_ = 0.44 mg/mL). In turn, we did not observe α-amylase inhibitory activity for hydrolyzates obtained from millet after the 65 °C treatment, except for the albumin fraction. This indicated that this temperature used for millet preparation caused release of peptides with potential α-amylase inhibitory activity only during albumin digestion. The highest IC_50_ values were noted for the fraction obtained from glutelin after the treatment with 100 °C (0.12 mg/mL); this value was higher than in the control sample (1.38 mg/mL). Moreover, the highest potential α-glucosidase inhibitory activity was noted for all samples obtained from millet after the 65 °C treatment compared with control samples. The lowest IC_50_ value was determined for the prolamin hydrolyzate (0.06 mg/mL).

All hydrolyzates were fractionated into peptide fractions with molecular mass under 3.0 kDa and the potential inhibitory activity towards enzymes involved in metabolic syndrome was determined, as shown in Table 3. It should be noted that, compared with the hydrolyzates, not all activities were determined for the peptide fractions. The peptide fraction obtained from prolamin after the 100 °C treatment was characterized by the highest ACE inhibitory activity (IC_50_ = 0.33 mg/mL). No α-amylase inhibitory activity was observed for all peptide fractions obtained from millet protein after the 65 °C treatment. Moreover, only the peptide fraction obtained from albumin hydrolyzates after the 100 °C treatment exhibited this activity (IC_50_ = 0.24 mg/mL). In turn, all control samples had α-amylase inhibitory activity, with the highest value noted for the peptide fraction obtained from control prolamin (0.11 mg/mL). Not all of the peptide fractions tested exhibited α-glucosidase inhibitory activity, and the highest values were noted for the albumin and globulin 11S peptide fractions obtained from millet treated with 65 °C, where the IC_50_ parameter had the significantly lowest value (0.05 mg/mL).

### 3.3. Effect of Protein Hydrolyzates and Peptide Fractions on HECa 10 Cells

To determine the influence of the protein hydrolyzates and peptide fractions on HECa 10 cells, samples (prolamin control, globulin 11S 65 °C, and prolamin 100 °C) with the highest potential inhibitory activity were chosen. Since the results varied, at least two of the highest inhibitory activity values from each of the temperature values were assumed as a criterion. Therefore, the prolamin control among the control samples was characterized by the highest ACE and α-amylase activity; globulin 11S 65 °C among samples obtained after 65 °C grain heated was characterized by the highest ACE and α-glycosidase activity; prolamin 100 °C among samples obtained after heating the grain at 100 °C was characterized by the highest ACE and α-glycosidase activity.

#### 3.3.1. Effect of the Prolamin Hydrolyzate (PRO H) and Peptide Fraction (PRO P) from Control Prolamin–MTT and NR Tests

Both tests used in the study (MTT and NR) measured different metabolic parameters of cells for determination of cell viability. The MTT assay measured cell metabolic activity by reduction of the tetrazolium dye MTT to its insoluble formazan form through nicotinamide adenine dinucleotide phosphate (NADPH)-dependent cellular oxidoreductase enzymes [24]. In contrast, in the NR assay, cell viability is determined by cellular uptake of neutral red dye (NR) incorporated in their lysosomes [25]. Some slight differences in the cell viability after the protein treatment observed in the MTT and NR tests may result from different metabolic parameters of cells measured by these tests or modulation of cell metabolism by the studied compounds (i.e., effect of the metabolic state of the cell).

In the proliferation assay (MTT and NR tests), both PRO H and PRO P samples were characterized by similar results in the range of the concentrations tested. Low concentrations of both protein hydrolyzates (0.1 and 1 μg/mL) resulted in a significant reduction in the number of cells after 24 h incubation. Higher concentrations of the tested protein fractions (50 and 100 μg/mL) caused an increase in the number of cells, as shown in Figure 1.

In both tests (MTT and NR), the concentrations of 5 and 10 μg/mL were the cut-off point between the negative and positive effects of the fractions on the number of cells. There were no significant changes in the percentage of live cells in both samples tested. The observed changes in the number of cells were associated with activation of apoptosis. The addition of PRO H at a concentration of 0.1 and 10 μg/mL resulted in the highest increase in the percentage of apoptotic cells in all the concentrations studied, compared to the control conditions. At the concentration of 100 μg/mL, there were no significant differences in the number of apoptotic cells, compared to the control. Interestingly, the low concentration of the protein hydrolyzate (0.1 and 10 μg/mL) showed anti-necrotic activity. In contrast, PRO P caused an increase in apoptosis at all the concentrations tested (0.1, 10, and 100 μg/mL). There was no effect of this sample on the percentage of necrotic cells. These changes are presented in Figure 2.

The addition of PRO H for 24 h to the HECa10 cells did not affect the cell cycle phases, as in the case of the addition of 0.1 and 10 μg/mL of PRO P. In contrast, a significant reduction in the number of cells in the G2 phase was noted at the concentration of 100 μg/mL of PRO P. This may indicate a reduction in the time required for the mitotic division, which could somehow explain the results obtained in cytotoxicity studies (a small but significant increase in the cell counts), as shown in Figure 3.

#### 3.3.2. Effect of the Hydrolyzate (G11S H) and Peptide Fraction (G11S P) from Globulin Obtained from Millet in the 65 °C Treatment

Incubation of HECa10 cells with G11S H (24 h) did not show significant changes in the number of cells measured by MTT and NR assays. Similar results were obtained for the G11S P in the NR test (no changes). However, in the MTT test for G11S P, a slight but significant increase in the cell counts was demonstrated, compared to the values obtained for the control conditions. The certain discrepancy between the results obtained for the G11S P in the MTT and NR test may be related to the characteristics of the test. Both assays measure the relative number of cells in different ways: the MTT test measures the metabolic activity of cells and the NR test assesses binding of dye to lysosomes. The slight differences observed in the MTT assays between the control and experimental conditions and the lack of an effect in NR prompt a conclusion that there is no significant effect of the G11S P peptide fraction on the number of cells, as shown in Figure 4. This observation was confirmed in the analysis of cell viability. There were no significant changes in the percentage of live, as shown in Figure 5, apoptotic, and necrotic cells after the 24 h incubation with the peptide fraction G11S P. The incubation with G11S H caused a significant reduction in the percentage of live cells, combined with an increase in the percentage of apoptotic cells (cells mainly in the early phase of apoptosis). The other tested concentrations did not cause significant changes in cell viability.

Moreover, there were no significant changes in the individual phases of the cell cycle after the 24 h incubation with the tested samples, as shown in Figure 6.

#### 3.3.3. Effect of the Hydrolyzate (PH) and Peptide Fraction (PP) from Prolamin after the 100 °C Treatment

There were no significant changes in the number of cells measured by MTT and NR tests for both the heat-treated millet hydrolyzate and peptide fractions (PH and PP) in the range of tested concentrations (0.1–100 μg/mL), except for the NR test and PP (concentration 100 μg/mL), where a statistically significant decrease in the cell count was observed, compared to the control values, as shown in Figure 7. Both samples obtained from the thermally-treated millet increased the percentage of apoptotic cells at 100 μg/mL (PH and PP). There was also a significant increase in the percentage of apoptotic cells for the PP peptide fraction at a concentration of 10 μg/mL. These changes in the percentage of apoptotic cells for the PP peptide fraction also resulted in a reduction in the percentage of live cells, as shown in Figure 8. The results did not show any significant changes in the cell cycle phases after the 24 h incubation with the tested PH and PP samples, as shown in Figure 9.

### 3.4. Characteristics and Identification of Peptide Fractions with Molecular Mass Under 3.0 kDa with the Highest Potential Inhibitory Activity Towards Enzymes Involved in Metabolic Syndrome

Peptide fractions PRO P, G11S P, and PP with the highest potential inhibitory activity towards enzymes involved in metabolic syndrome were separated on Sephadex G10, as shown in Figure 10. After this process, all peptide fractions were separated into two fractions (1 and 2 for PRO P, I and II for G11S P; A and B for PP). The ACE, α-amylase, and α-glucosidase inhibitory activity was investigated in these fractions, as shown in Table 4. The highest potential for inhibition of metabolic syndrome was determined in the first fraction obtained from PRO P demonstrating the highest activity, where the IC_50_ values for ACE, α-amylase, and α-glucosidase were 4.82, 43.56, and 91.38 µg/mL, respectively. Also, the first fraction obtained from G11S P showed the highest properties (IC_50_ = 10.01, 60.71, and 35.06 µg/mL for ACE, α-amylase, and α-glucosidase, respectively). However, the second fraction from PP was characterized by the highest potential enzyme inhibitory activity with IC_50_ values of 31.27, 118.12, and 117.01 µg/mL for ACE, α-amylase, and α-glucosidase, respectively. Therefore, fractions 1, II, and B were selected for identification of peptide sequences with the LC-MS-MS/MS technique.

The peptide sequences were identified as GQLGEHGGAGMG and GEHGGAGMGGGQFQPV from fraction 1, EQGFLPGPEESGR and RLARAGLAQ from fraction II, and YGNPVGGVGH and GNPVGGVGHGTTGT from fraction B. These sequences were derived from the uncharacterized protein of *Panicum hallii*.

## 4. Discussion

Different food processing techniques, e.g., fortification, soaking, cooking, germination, or fermentation, are used to improve the quality of nutrient compounds as well as the digestibility and bioavailability of food nutrients while reducing anti-nutrients [26,27]. In this study, we investigated the potential antidiabetic and antihypertensive activity of protein hydrolyzates and peptide fractions obtained from millet after treatment at different temperature values. Moreover, the effect of peptide fractions with the highest antidiabetic and antihypertensive activity on endothelial cells was determined. Bioactive substances obtained from the digestion of the food affect metabolism of various cells types, which mainly applies to intestinal epithelial and endothelial cells. It has been repeatedly shown that food substances regulate the proliferation, differentiation, and regulation of intestinal epithelial cells [28]. This has also been demonstrated for endothelial cells [29].

The main parameter determining the biological quality of food protein is the degree of hydrolysis, because it affects the potential bioavailability of its peptides and amino acids [30]. It should be noted that not only proteases but also α-amylase were used in the hydrolysis model in this study. The latter is the first active digestion enzyme in the gastrointestinal tract. According to the data described in Table 1, DH was also noted after hydrolysis of α-amylase. This indicates that the proteins may be combined with sugar glycosidic bonds and, using the amylase enzyme, release proteins from complexes, making them more available to proteases. Similar results were obtained in our previous study, where bean proteins were hydrolyzed [23]. The highest DH values were noted for hydrolyzates obtained after the last step of the process (34.47–98.33%). The digestion of millet proteins in in vitro conditions by gastrointestinal enzymes was found to be effective, compared to the DH values of rice bran protein fractions: albumin, globulin, prolamin, and glutelin, where this factor was in the range of 9–22% [31]. Generally, we did not observe a significant relationship between the temperature of millet preparation and the DH values. This parameter depends on the protein fraction. The DHs for albumins and globulins 11S obtained from the millet treatment at 65 °C and 100 °C had lower values, compared with the control samples. Data reported by Pan et al. [32] on the change in DH during the hydrolysis of the lotus seed protein pretreated at different temperatures (50–80 °C) indicated that the DHs of lotus seed protein pretreated at 50 and 60 °C were significantly higher, while the DHs of lotus seed protein pretreated at 70 and 80 °C were lower, compared with the control sample (non-pretreated). The differences presented in the data may result from the use of different protein sources, temperatures, and enzymes. Protein PAC and PAV are other important parameters that influence the potential activity of protein. The peptide bioaccessibility index of all samples was higher than 1, which indicates that the peptides released from the millet protein fraction were highly bioaccessible in vitro, as shown in Table 1. A positive effect of the temperature treatment on this factor was observed in the case of globulin 11S, prolamin, and glutelin. It should be noted that a higher difference compared with the control sample was determined for the globulin 11S treatment at 100 °C, where the value of this parameter was 23.89. On the other hand, PAV indicated that the peptides were generally poorly bioavailable in vitro, except the peptides obtained from globulin 11S and prolamin treated at 65 °C, where the PAV values were higher than 1 (2.12 and 1.24, respectively). This suggests that the use of a lower temperature than that which is traditionally used for preparation of millet can increase the bioavailability of peptides. We observed that the biological activity of proteins and peptides obtained from millet grains subjected to the temperature treatment was different. Generally, the hydrolyzates and peptide fractions from the prolamin control had an effect on the proliferation of the HECa10 cells (low concentration: 0.1–1 µM decreased and high concentration: 50–100 µM increased the proliferation), as shown in Figure 1. After the high-temperature treatment (100 °C), the prolamin hydrolyzates did not exhibit these properties, as shown in Figure 4. Similarly, the hydrolyzates obtained from globulin 11S derived from millet after the 65 °C treatment did not have an impact on the cell number, as shown in Figure 7. However, some biological properties of the hydrolyzates and peptide fractions from prolamin (i.e., induction of apoptosis) were unaffected by the heat treatment. These changes caused by temperature may be associated with several heat-induced modifications such as bioavailability of active peptides for digestive enzymes, thermal treatment time, or peptide aggregation via formation of covalent or non-covalent bonds [33,34]. In summary, it seems that increased temperature may be a good method for increasing the bioaccessibility and reduction of the negative impact of obtained hydrolyzates on cells. However, it should be noted that only the protein fraction of millet grains was analyzed in the present study. For other biologically active compounds, for example phenolics, enhanced antioxidant activity and anti-proliferative properties were found after the high temperature treatment (steam flash explosion) [35].

There are several studies indicating that millet grain could be a beneficial food component in obesity-related diseases such as type 2 diabetes and cardiovascular diseases [36]. The in vitro antidiabetic effects of the protein hydrolyzates in this study were determined by analysis of α-glucosidase and α-amylase inhibitory activity, as shown in Table 2. There are two main enzymes involved in starch digestion and glucose release. One of them, i.e., α-glucosidase, hydrolyzes starch to sugar fragments that are converted into glucose by α-amylase. Therefore, the inhibition of α-glucosidase activity can effectively reduce the production of glucose from starchy foods and decrease the level of blood sugar in diabetic patients [33]. It should be noted that almost all of the hydrolyzates obtained from the temperature-treated millet were characterized by a lower IC_50_ value than samples without the temperature treatment. In the case of glutelin, the application of heat to prepare millet resulted in the appearance of the α-glucosidase inhibitory activity, whereas this activity was not detected in the control sample. Moreover, the ACE inhibitory activity of the hydrolyzates was found to exert a potential antihypertensive effect. All of the tested samples were characterized by lower IC_50_ values than the control, which indicated that the temperature used for millet preparation contributed to improvement of these properties.

Metabolic syndrome and related diseases (obesity, diabetes, arteriosclerosis) are associated with endothelial cell dysfunction [37]. This applies to both normal cell metabolism (i.e., proliferation, differentiation, apoptosis) and their functions (i.e., cytokine and chemokine secretion) [38]. Repair of endothelial cell dysfunctions is a new strategy in metabolic syndrome treatment in humans [39]. Therefore, in the present work, we check whether the digested hydrolyzates can modulate cell metabolism: proliferation, viability, or modification of the cell cycle. Prolamin obtained by the hydrolysis of non-heat treated millet grains at the low concentration (0.1–1 µM) decreased the number of endothelial HECa10 cells, as shown in Figure 1, which was associated with an increased apoptosis and decreased necrosis level, as shown in Figure 2. The higher concentration of prolamins normalized the apoptosis level, which caused an increase in the cell number in comparison to the control, as shown in Figure 1 and Figure 2. Prolamin obtained from millet grains treated with 100 °C did not show anti-proliferation activity, as shown in Figure 7, which was associated with the normalized level of apoptosis at the low concentration, as shown in Figure 5. Interestingly, at the highest concentration tested (100 µM), a significant increase in apoptosis was found with a slight, but not significant, reduction of necrosis, as shown in Figure 8. In both the temperature-treated or untreated millet grains, there were no significant changes in the cell cycle, as shown in Figure 3 and Figure 9. Similarly to the temperature-affected prolamin, globulin 11S did not influence the proliferation, viability, and cell cycle in the tested concentrations, except for the highest concentration, in which a reduced percentage of viable cells and an increased percentage of apoptotic cells were found, as shown in Figure 4, Figure 5 and Figure 6. The results indicated that proteins obtained from millet grains are safer for endothelial cell metabolism after high-temperature pretreatment.

Since peptides may exhibit their properties after penetration to blood [40], we fractionated the hydrolyzates into peptide fractions with molecular mass under 3.0 kDa. The peptide fractions were characterized by lower IC_50_ values than the hydrolyzates. This result corresponds well with that described by Uraipong and Zhao [31], where peptide fractions with molecular mass under 3.0 obtained from rice bran protein had higher α-glucosidase inhibitory activity than crude digests. It should be noted that the α-glucosidase inhibitory activity in this study was noted not for all the peptide fractions tested. We did not observe this activity in the case of prolamin control and globulin 11S 65 °C, while it was noted for their hydrolyzates. The biological activity of peptides tested on the endothelial HECa10 cell line showed some slight differences in comparison to hydrolyzates. The prolamin peptides from millet grains prepared in standard conditions exhibited an increased level of apoptosis at the highest concentration (100 µM), in which the cell cycle was modified (reduced percentage of cells in the G2 phase), as shown in Figure 2 and Figure 3. Congruous results of cell viability were found for prolamin peptides from millet grains pretreated with high temperature (increased apoptosis and decreased viability at the 10 and 100 µM concentrations), as shown in Figure 8. Peptides from globulins 11S did not significantly affect all tested parameters of the cells, as shown in Figure 4, Figure 5 and Figure 6. These results may indicate that the activity of peptides with molecular mass higher than 3.0 kDa or the peptides in the hydrolyzates acted synergistically [41]. Among patients with type 1 or type 2 diabetes, cardiovascular diseases are one of the causes of morbidity and mortality. However, the exact mechanisms linking the development of atherosclerosis and cardiovascular disease among diabetics are still poorly understood. It is suggested that there is a relationship between hyperglycemia and intracellular metabolic modification induced by oxidative stress, inflammation process, and endothelial dysfunction [42,43].

The peptides with physiological activity in the organism comprised 2–20 amino acids, and this length of molecules allows them to cross the intestinal barrier to exert their effect at the tissue level [44]. There are many studies that indicate that ACE inhibitors contain aromatic residues such as proline, tryptophan, or phenylalanine in the C-terminal tripeptide sequence or aliphatic amino acids such as glycine, valine, leucine, and isoleucine at the N-terminal position [45]. The relationship between the peptide structure and biological activity has not been fully established. Some longer peptides exhibit high ACE inhibitory activity while those including hydrophilic amino acid residues are characterized by low inhibitory activities [46]. This activity may also be related to the presence of hydrophilic amino acid residues in the structure of ACE inhibitors, which determine peptide binding to the enzyme active site [47]. Moreover, the results obtained by Ngoh et al. [48] indicated that strong α-amylase inhibition was induced by histidine, proline, and methionine present in the structure of peptides. Also, the structure of peptides with α-glucosidase inhibitory activity is not clear due to the fact that many inhibitors are described as complex and the relationship between their structure and activities is difficult to determine. A peptide with QPGR sequences that can be a potential drug for diabetes treatment was identified from silkworm pupae [49] and the N-terminal sequence GR was in agreement with the results obtained in our study.

## 5. Conclusions

Proteins derived from food may not only serve a regulatory function and be a source of energy but also be precursors of peptides or amino acids. However, the effect of these compounds is associated not only with their source but also with the way of food preparation. Peptides are involved in different physiological and regulatory functions in the organism. Although there are studies about the influence of peptides on inhibition of metabolic syndrome pathogenesis, the mechanisms by which peptides or protein hydrolyzates control glucose levels are still poorly understood. Thus, these results provide some data about the effect of protein hydrolyzates and peptide fractions on the activity of enzymes involved in metabolic disorders. The results of this study showed that the temperature treatment of millet grains led to different degrees of protein hydrolysis in in vitro conditions and the resultant peptides were characterized by high α-amylase, α-glucosidase, and ACE inhibitory activities. The prolamin from the control grains and from grains subjected to the 100 °C treatment as well as globulin 11S from grains after the 65 °C treatment had an especially strong effect on the metabolic syndrome enzymes. The tests performed on endothelial cells indicated that the heat pre-treatment of the millet grains attenuated the negative impact of the digested hydrolyzates and the prolamin and globulin 11S peptide fractions. Despite the effect of the hydrolyzates and peptide fractions on the HECa10 cells, it should be noted that we did not observe a clear relationship between the concentrations of the samples and their biological effect (especially in the temperature-treated grains). The sequences of potential inhibitory peptides were identified as GEHGGAGMGGGQFQPV, EQGFLPGPEESGR, RLARAGLAQ, YGNPVGGVGH, and GNPVGGVGHGTTGT.

Nevertheless, these results are a contribution to further research which should be undertaken to verify the findings in in vivo studies using animal models and human clinical trials. Moreover, the results can be the basis for further research on other species of millet or cereals.

## Figures and Tables

**Figure 1 nutrients-11-00550-f001:**
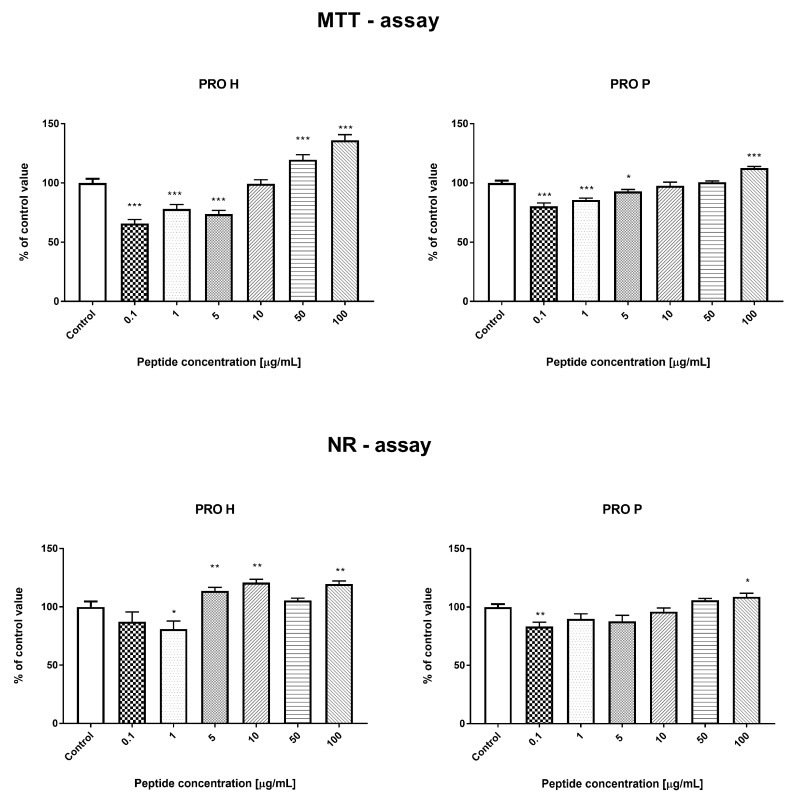
Result of MTT and neutral red (NR) tests for prolamin hydrolyzate (PRO H) and peptide fraction (PRO P). * *p* < 0.05; ** *p* < 0.01; *** *p* < 0.001.

**Figure 2 nutrients-11-00550-f002:**
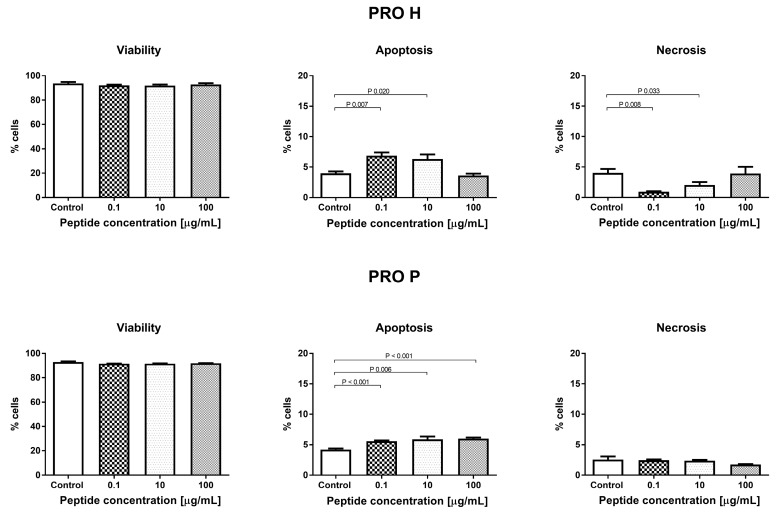
Evaluation of cell apoptosis, necrosis, and viability with PRO H and PRO P.

**Figure 3 nutrients-11-00550-f003:**
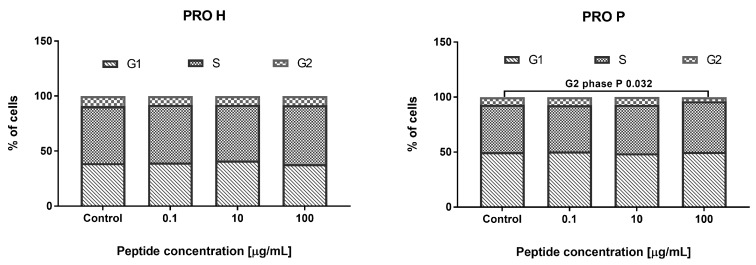
The effect of PRO H and PRO P on the cell cycle phases.

**Figure 4 nutrients-11-00550-f004:**
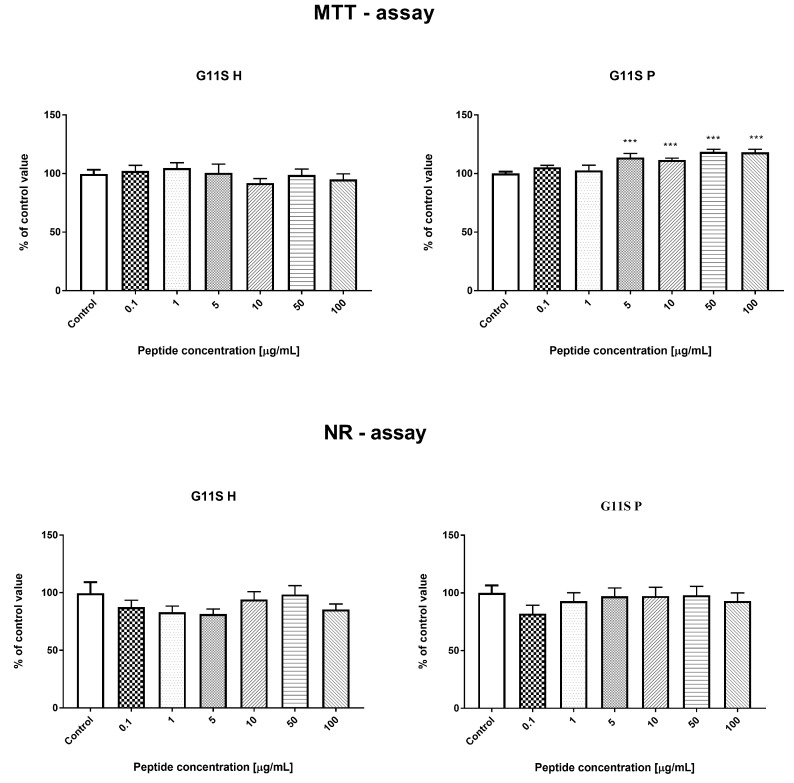
Result of MTT and NR tests for globulin hydrolyzate (G11S H) and globulin peptide fraction (G11 P). *** *p* < 0.001.

**Figure 5 nutrients-11-00550-f005:**
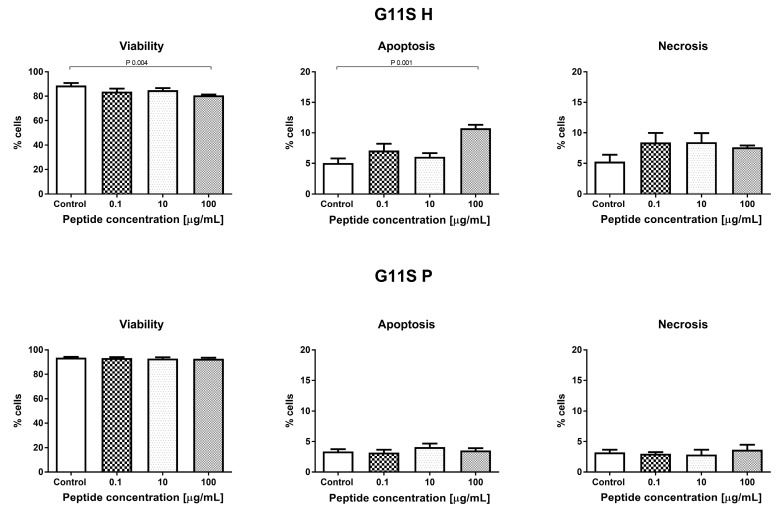
Evaluation of cell apoptosis, necrosis, and viability with G11S H and G11S P.

**Figure 6 nutrients-11-00550-f006:**
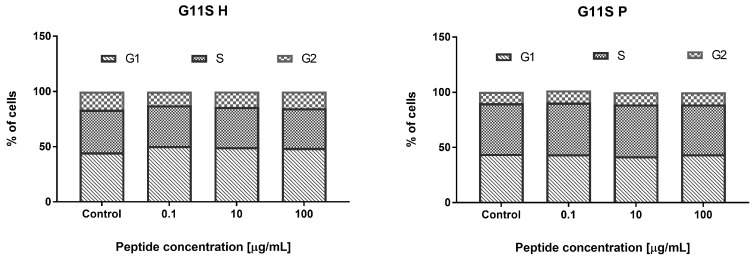
The effect of G11S H and G11S P on the cell cycle phases.

**Figure 7 nutrients-11-00550-f007:**
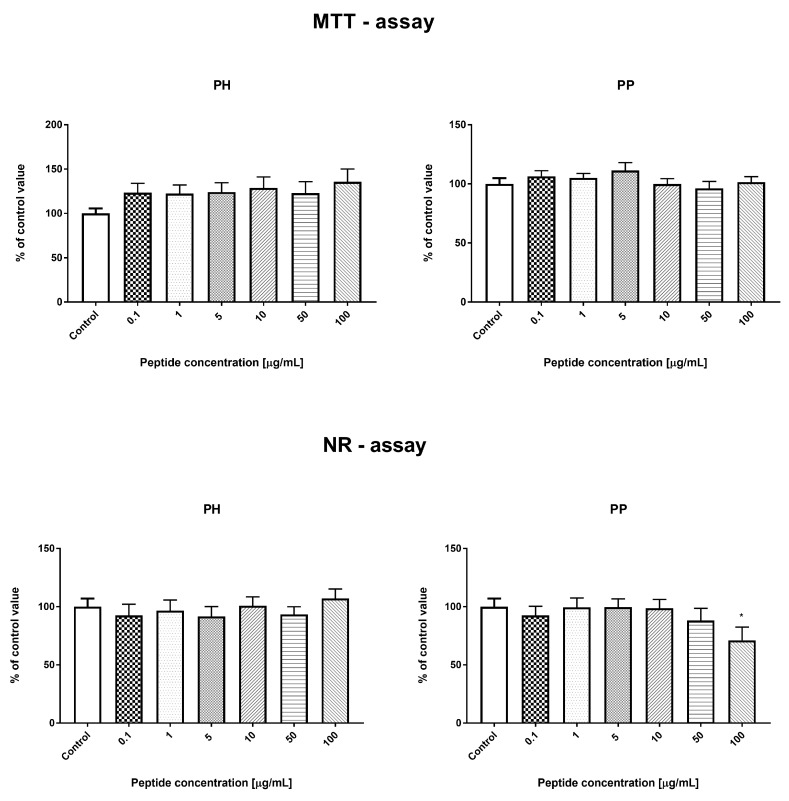
Result of MTT and NR tests for prolamin hydrolyzate (PH) and prolamin peptide fraction (PP). * *p* < 0.05.

**Figure 8 nutrients-11-00550-f008:**
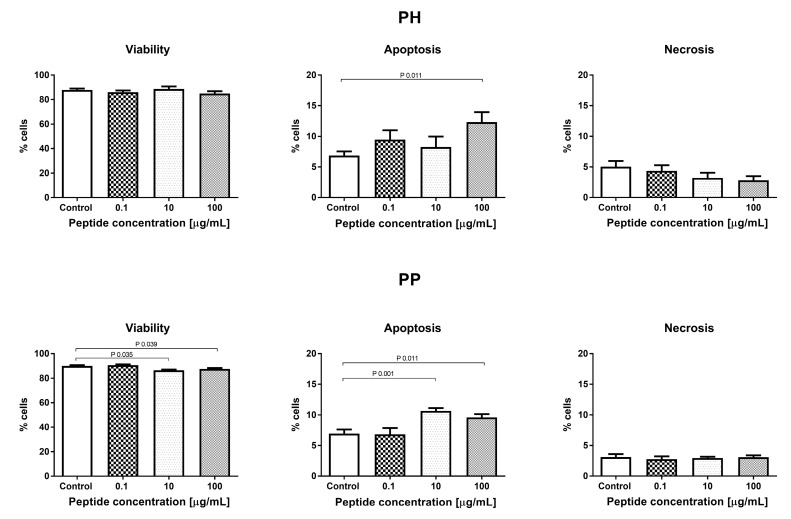
Evaluation of cell apoptosis, necrosis, and viability with PH and PP.

**Figure 9 nutrients-11-00550-f009:**
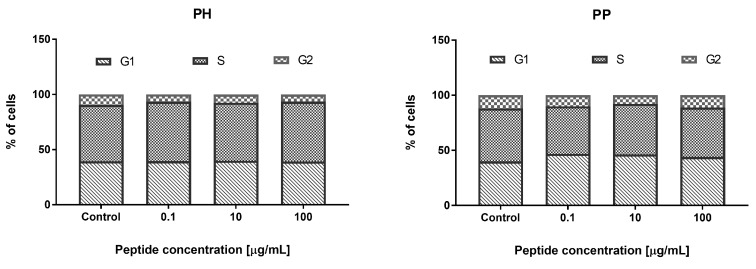
The effect of PH and PP on the cell cycle phases.

**Figure 10 nutrients-11-00550-f010:**
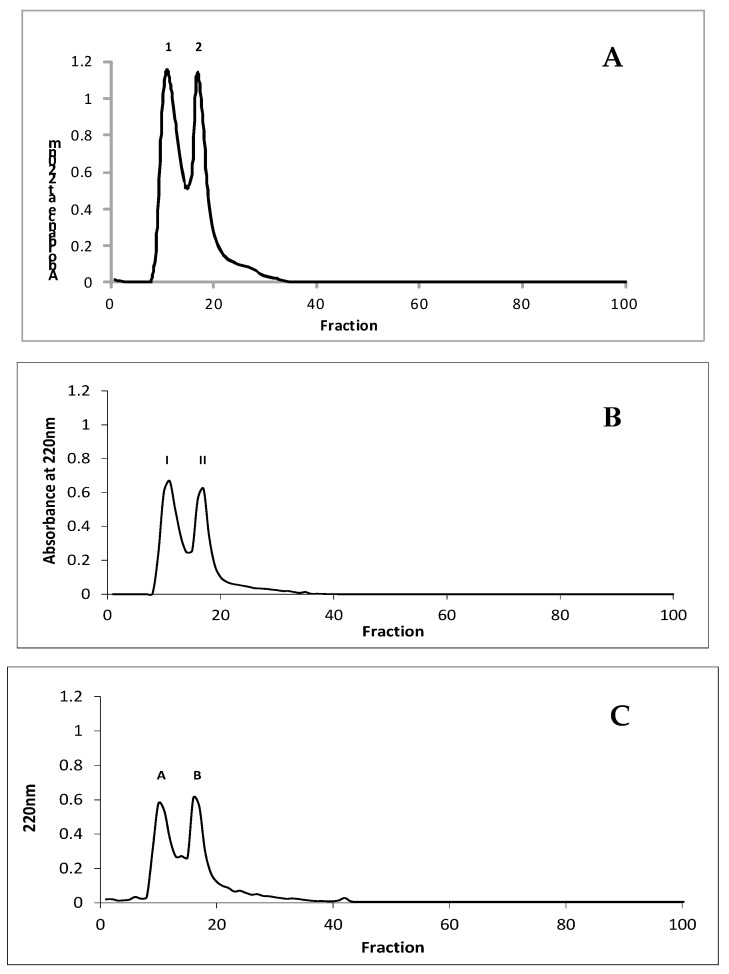
Peptides’ profile separated on Sephadex G10 (PRO P—**A**, G11S P—**B**, and PP—**C**).

**Table 1 nutrients-11-00550-t001:** Degree of hydrolysis (DH) (%) of protein fractions during hydrolysis under gastrointestinal conditions and potential bioaccessibility (PAC) and potential bioavailability (PAV) factors.

	Enzyme	α-Amylase	Pepsin	Pancreatin	PAC	PAV
Protein						
65 °C						
Albumin		69.24 ± 1.32 ^Aa^	85.36 ± 1.74 ^Ba^	91.52 ± 1.55 ^CAb^	1.06	0.12
Globulin 7S		59.99 ± 1.22 ^Aa^	63.03 ± 1.61 ^Aa^	86.69 ± 1.47 ^Ba^	1.96	0.40
Globulin 11S		26.46 ± 0.89 ^Aa^	30.81 ± 1.11 ^Ba^	34.47 ± 1.01 ^Ca^	6.47	2.12
Prolamin		48.67 ± 2.01 ^Aa^	53.77 ± 1.37 ^Ba^	62.23 ± 1.87 ^Ca^	2.74	1.24
Glutelin		41.46 ± 1.44 ^Aa^	43.21 ± 1.78 ^Aa^	57.09 ± 1.31 ^Ba^	1.94	0.27
100 °C						
Albumin		56.31 ± 2.17 ^Ab^	76.16 ± 1.57 ^Bb^	89.20 ± 2.07 ^Ca^	1.14	0.14
Globulin 7S		29.26 ± 0.69 ^Ab^	30.29 ± 0.77 ^Ab^	46.26 ± 1.69 ^Bb^	2.92	0.36
Globulin 11S		13.78 ± 0.98 ^Ab^	20.47 ± 0.66 ^Bb^	47.68 ± 1.78 ^Cb^	23.89	0.57
Prolamin		13.19 ± 0.36 ^Ab^	20.51 ± 0.91 ^Bb^	47.52 ± 1.21 ^Cb^	9.56	0.56
Glutelin		62.34 ± 1.25 ^Ab^	64.59 ± 1.24 ^Ab^	88.47 ± 1.74 ^Bb^	3.73	0.24
C						
Albumin		54.00 ± 1.88 ^Ab^	62.04 ± 1.70 ^Bc^	94.45 ± 2.07 ^Cb^	1.92	0.11
Globulin 7S		66.89 ± 1.98 ^Ac^	78.47 ± 2.00 ^Ac^	98.33 ± 1.21 ^Ac^	1.18	0.15
Globulin 11S		34.51 ± 1.14 ^Ac^	38.07 ± 0.77 ^Bc^	38.56 ± 1.87 ^Bc^	1.18	0.21
Prolamin		46.56 ± 1.22 ^Aa^	48.69 ± 1.01 ^Ac^	57.82 ± 1.29 ^Bc^	1.34	0.15
Glutelin		58.08 ± 1.63 ^Ac^	60.12 ± 1.44 ^Ac^	81.26 ± 1.89 ^Bc^	1.79	0.14

C—grains without heating. All values are mean ± standard deviation for triplicate experiments. Different lower case letters in the same protein fraction at the different temperature s used indicate a significant difference (α = 0.05). Different capital letters at the same enzyme temperatures used indicate a significant difference (α = 0.05).

**Table 2 nutrients-11-00550-t002:** The IC_50_ values (mg/mL) of hydrolyzates for inhibitory activity of enzymes involved in metabolic syndrome pathogenesis.

	Temperature	65 °C	100 °C	C
Protein				
		ACE		
Albumin		3.25 ± 0.87 ^ABa^	2.40 ± 0.74 ^Aab^	4.73 ± 0.54 ^Ba^
Globulin 7S		2.00 ± 0.01 ^Ab^	2.63 ± 0.36 ^Aa^	4.85 ± 0.44 ^Ba^
Globulin 11S		0.44 ± 0.01 ^Ac^	1.50 ± 0.02 ^Bbc^	4.89 ± 0.21 ^Ca^
Prolamin		1.24 ± 0.03 ^Ab^	1.38 ± 0.01 ^Ac^	3.52 ± 0.31 ^Bb^
Glutelin		1.73 ± 0.12 ^Ab^	2.12 ± 0.17 ^Aabc^	6.39 ± 0.28 ^Bc^
		α-amylase		
Albumin		1.37 ± 0.02 ^A^	3.84 ± 0.34 ^Ba^	1.92 ± 0.01 ^Ca^
Globulin 7S		nd	5.47 ± 0.11 ^Ab^	3.27 ± 0.51 ^Bb^
Globulin 11S		nd	2.37 ± 0.08 ^Aa^	6.32 ± 0.44 ^Bc^
Prolamin		nd	8.21 ± 1.17 ^Ac^	0.77 ± 0.01 ^Bd^
Glutelin		nd	0.12 ± 0.01 ^Ad^	1.38 ±0.04 ^Bad^
		α-glucosidase		
Albumin		0.08 ± 0.001 ^Aa^	0.60 ± 0.013 ^Ba^	0.49 ± 0.011 ^Ca^
Globulin 7S		0.58 ± 0.002 ^Ab^	0.89 ± 0.017 ^Bb^	1.46 ± 0.021 ^Cb^
Globulin 11S		0.24 ± 0.003 ^Ac^	0.35 ± 0.018 ^Bc^	0.12 ± 0.001 ^Cc^
Prolamin		0.06 ± 0.003 ^Ad^	0.51 ± 0.013 ^Bd^	1.13 ± 0.001 ^Cd^
Glutelin		0.57 ± 0.012 ^Ab^	0.60 ± 0.015 ^Aa^	nd

C—grains without heating. ACE— angiotensin-converting enzyme. nd—activity not detected. All values are mean ± standard deviation for triplicate experiments. Different capital case letters in the same protein fraction indicate a significant difference (α = 0.05). Different lower case letters in the same enzyme temperature used indicate a significant difference (α = 0.05).

**Table 3 nutrients-11-00550-t003:** The IC_50_ (mg/mL) values of peptide fractions on activity of enzymes involved in metabolic syndrome pathogenesis.

	Temperature	65 °C	100 °C	C
Protein				
		ACE		
Albumin		0.41 ± 0.002 ^Aa^	0.45 ± 0.012 ^Aa^	0.83 ± 0.011 ^Ba^
Globulin 7S		0.54 ± 0.013 ^Ab^	0.37 ± 0.011 ^Bb^	0.60 ± 0.022 ^Cb^
Globulin 11S		0.38 ± 0.015 ^Aa^	0.65 ± 0.021 ^Bc^	0.79 ± 0.025 ^Ca^
Prolamin		0.54 ± 0.016 ^Ab^	0.33 ± 0.001 ^Bd^	0.42 ± 0.010 ^ABc^
Glutelin		0.66 ± 0.018 ^Ac^	0.63 ± 0.014 ^ABc^	0.61 ± 0.012 ^Bb^
		α-amylase		
Albumin		nd	0.24 ± 0.014 ^A^	0.39 ± 0.017 ^Ba^
Globulin 7S		nd	nd	0.30 ± 0.001 ^b^
Globulin 11S		nd	nd	0.44 ± 0.011 ^c^
Prolamin		nd	nd	0.11 ± 0.002 ^d^
Glutelin		nd	nd	0.67 ± 0.012 ^e^
		α-glucosidase		
Albumin		0.05 ± 0.004 ^Aa^	0.26 ± 0.022 ^Ba^	0.10 ± 0.001 ^Ca^
Globulin 7S		0.22 ± 0.001 ^Ab^	0.29 ± 0.002 ^Bb^	0.14 ± 0.001 ^Cb^
Globulin 11S		0.05 ± 0.001 ^Aa^	nd	0.09 ± 0.013 ^Ba^
Prolamin		0.18 ± 0.003 ^Ac^	0.12 ± 0.001 ^Ba^	nd
Glutelin		0.06 ± 0.003 ^Ad^	0.31 ± 0.002 ^Bd^	nd

C—grains without heating. nd—activity not detected. All values are mean ± standard deviation for triplicate experiments. Different capital case letters in the same protein fraction indicate a significant difference (α = 0.05). Different lower case letters in the same enzyme temperature used indicate a significant difference (α = 0.05).

**Table 4 nutrients-11-00550-t004:** IC_50_ (µg/mL) of peptide fractions for inhibitory activity of enzymes involved in metabolic syndrome pathogenesis.

Sample	ACE	α-Amylase	α-Glucosidase
Fraction from PRO P:			
1	4.82 ± 0.13 ^Aa^	43.56 ± 4.75 ^Ba^	91.38 ± 5.37 ^Ca^
2	23.61 ± 1.18 ^Ab^	49.73 ± 2.59 ^Bb^	87.69 ± 1.83 ^Ca^
Fraction from G11S P:			
I	10.01 ± 0.51 ^Ac^	60.71 ± 5.05 ^Bc^	35.06 ± 6.03 ^Cb^
II	13.28 ± 0.16 ^Ad^	68.74 ± 2.09 ^Bb^	36.20 ± 3.11 ^Cb^
Fraction from PP:			
A	82.02 ± 2.01 ^Ae^	127.96 ± 9.82 ^Bd^	128.38 ± 2.51 ^Bc^
B	31.27 ± 1.17 ^Af^	118.12 ±1.65 ^Be^	107.01 ± 2.22 ^Bd^

All values are mean ± standard deviation for triplicate experiments. Different capital case letters in the same peptide fraction indicate significant difference (α = 0.05). Different lower letters for the same enzyme indicate significant difference (α = 0.05).

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
