# Peer review of "Different Temperature Treatments of Millet Grains Affect the Biological Activity of Protein Hydrolyzates and Peptide Fractions"

_nutrients, 2019, doi:10.3390/nu11030550_

Reviewer 1 Report

The manuscript entitled “The effect of temperature on the metabolic syndrome inhibitory activity and the influence of peptides derived from millet proteins on endothelial cell HECa10” (Manuscript ID: nutrients-438012) by Jakubczyk et al., investigates the effects of different protein fractions and peptides obtained from proso millet to inhibit several enzymes related to glucose absorption and hypertension as well as on the viability of HECa10 cells. The actions of different temperatures on these fractions were also tested. This study is in line with a previous study from the same authors focused on peptides obtained from pea proteins.

Some of my concerns are the following:

1.- The title is a bit messy and does not provide a clear idea of the objective or main findings of the study

2.- The introduction is too long. In addition and more importantly, the authors state in the introduction section and in the discussion part of the manuscript that α-amylase and α-glucosidase are target enzymes for diabetes in the context of metabolic syndrome. I agree with the fact that these enzymes are pivotal for glucose absorption; however, there are other critical enzymes in diabetes and metabolic syndrome. Since metabolic syndrome is usually accompanied by obesity, other enzymes involved in lipogenesis or lipolysis should be analyzed and other cell lines such as enterocytes or adipocytes should be used apart from HECa10 cells to be able to obtain robust conclusions. Finally, as metabolic syndrome is usually associated with type 2 diabetes and insulin resistance, more experiments concerning insulin sensitivity (actions of these protein fractions and peptides on insulin sensitivity in cultured adipocytes, for example) should be included to reinforce this part of the manuscript. In general, this study is a very preliminary chemical study that needs to be enlarger with more physiological experiments to be able to reach robust conclusions to extrapolate to humans. In case these experiments are not performed, the conclusions are limited and can only state that heat pretreatment is a good method for preparation of millet seeds to increase their in vitro bioaccesibility and bioavailability and also to increase their in vitro inhibitory actions on ACE, α-amylase and α-glucosidase activity.

3.- Please, rewrite the objective (lines 88-91). The authors have not tested different cooking processes, only two different temperatures (65 vs. 100ºC), but not frying, for example.

4.- The authors should justify why they have used the Panmicum liciaceum L (proso millet) and if the results obtained could be extrapolated to the other types of millet (pearl millet, foxtail millet etc.).

5.- The authors should justify why they have used different methods to measure cell viability (MTT and NR assays). What is the difference between them? They should also justify why the results are different depending the assays used. Finally, I don´t think MTT results should be presented as absorbances.

6.- If tables are split among two pages, please repeat the heading to make them clearer for the reader. In addition, a word is missing in foot Table 1. I think it should read as follows: Different capital letters in the same Enzyme? used temperature indicates…

7.- Line 275: please correct. Globulin 11S after 65ºC treatment (IC50= 0.44 mg/ml) and not 0.34…

8.- Lines 282 and 283: Please check … The lowest IC50 value was determined for the albumin hydrolyzate (0.01 mg/mL). I don´t see where this value come from.

9.- Results concerning cell cycle: please add the statistics along with the graphs.

10.- It is quite difficult to see that prolamin control, globulin 11S 65ºC and prolamin 100ºC are the samples with the highest inhibitory activity… Please, justify better and highlight the results if needed to make it clearer for the readers.

11.- Tables: I strongly suggest to use always Capital letters first and lower letters afterwards (in table 4 is just the opposite). In addition, I suggest to reduce the superscript letters (from a to f) in table 4.

12.- Discussion section:

420-423 and 484-487 are repetitive.

Avoid general conclusions or statements that have not been confirmed such as line 489: “… the temperature-treated millet could be used in inhibition of metabolic syndrome…”

Lines 506-512: this is not discussion but repetition of the results

13.- The in vitro results concerning PAC and PAV and enzymes are clear and summarized several times along the manuscript. However, the effects on HECa10 are not clear enough and should be summarized and highlighted.

 14.- Please check references. For example, check references 4, 8, 10, complete reference 25 (Food Chem 2013. 141 (4): 3774-80

15.- Line 424: correct “demonstrating”

16.- Please, write in vitro in italics along the manuscript

17.- Line 450: PAV does not need to be explained as it has already been explained before

18.- Line 455: please correct indication for indicating

19.- Delete peptide in line 515

Author Response

Thank you very much for giving us the opportunity to revise our manuscript. The title was change: Different temperature treatments of millet grains affect the biological activity of protein hydrolyzates and peptide fractions. We have tried our best to revise our manuscript according to the comments. The figures were changed. The reference list was carefully checked.

We look forward to hearing a positive reply.

Sincerely yours,

Anna Jakubczyk

The manuscript entitled “The effect of temperature on the metabolic syndrome inhibitory activity and the influence of peptides derived from millet proteins on endothelial cell HECa10” (Manuscript ID: nutrients-438012) by Jakubczyk et al., investigates the effects of different protein fractions and peptides obtained from proso millet to inhibit several enzymes related to glucose absorption and hypertension as well as on the viability of HECa10 cells. The actions of different temperatures on these fractions were also tested. This study is in line with a previous study from the same authors focused on peptides obtained from pea proteins.

 Some of my concerns are the following:

1.- The title is a bit messy and does not provide a clear idea of the objective or main findings of the study

The title was changed: Different temperature treatments of millet grains affect the biological activity of protein hydrolyzates and peptide fractions.

2.- The introduction is too long. In addition and more importantly, the authors state in the introduction section and in the discussion part of the manuscript that α-amylase and α-glucosidase are target enzymes for diabetes in the context of metabolic syndrome. I agree with the fact that these enzymes are pivotal for glucose absorption; however, there are other critical enzymes in diabetes and metabolic syndrome. Since metabolic syndrome is usually accompanied by obesity, other enzymes involved in lipogenesis or lipolysis should be analyzed and other cell lines such as enterocytes or adipocytes should be used apart from HECa10 cells to be able to obtain robust conclusions. Finally, as metabolic syndrome is usually associated with type 2 diabetes and insulin resistance, more experiments concerning insulin sensitivity (actions of these protein fractions and peptides on insulin sensitivity in cultured adipocytes, for example) should be included to reinforce this part of the manuscript. In general, this study is a very preliminary chemical study that needs to be enlarger with more physiological experiments to be able to reach robust conclusions to extrapolate to humans. In case these experiments are not performed, the conclusions are limited and can only state that heat pretreatment is a good method for preparation of millet seeds to increase their in vitro bioaccesibility and bioavailability and also to increase their in vitro inhibitory actions on ACE, α-amylase and α-glucosidase activity.

Thank you for your very interesting and inspiring comments that will be consider in our next research project. The introduction section was shortened. We agree with the fact that there are other enzymes that play role in metabolic syndrome and diabetes pathogenesis (e.g. dipeptidyl peptidase-4) but there are many studies about role of α-amylase and α-glucosidase inhibitors in insulin adjustment. The inhibition of these enzymes is a therapeutic target for retarding glucose absorption and sup-pressing postprandial hyperglycemia and thus can be potential anti-diabetic agent (Kunyanga et al., 2012; Ren et al., 2016; Abhishek et al., 2019). We also agree that metabolic syndrome is usually accompanied by obesity and the results about the effect of peptides on lipase activity and enzymes involved in inflammatory process was described in our study: Potential anti-inflammatory and lipase inhibitory peptides generated by in vitro gastrointestinal hydrolysis of heat treated millet grains – CyTa – Journal of Food (article in press).

References:

Abhishek, M., Somashekaraiah, B.V., Dharmesh, S.M. In vivo antidiabetic and antioxidant potential of Psychotria dalzellii in streptozotocin-induced diabetic rats. South African Journal of Botany 121 (2019) 494–499

Kunyangaa, C.K, Imungia, J.K, Okotha, M.W. , Biesalski H.K, Vadivel, V. Total phenolic content, antioxidant and antidiabetic properties of methanolic extract of raw and traditionally processed Kenyan indigenous food ingredients. LWT - Food Science and Technology 45 (2012) 269 – 276

Ren, Y., Liang, K.,  Jin, Y.,  Zhang, M., Chen Y., Wu H., Lai F. Identification and characterization of two novel α-glucosidase inhibitory oligopeptides from hemp (Cannabis sativa L.) seed protein. Journal of Functional Foods 26 (2016) 439–450

3.- Please, rewrite the objective (lines 88-91). The authors have not tested different cooking processes, only two different temperatures (65 vs. 100ºC), but not frying, for example.

It was corrected.

4.- The authors should justify why they have used the Panmicum liciaceum L (proso millet) and if the results obtained could be extrapolated to the other types of millet (pearl millet, foxtail millet etc.).

We have used Panicum miliaceum L. due to the fact that is one of the oldest cultivated and first domesticated crops. The other types of millet may have different protein profile so the results obtained in our study can be the basis for further research on other species of millet or cereals.

5.- The authors should justify why they have used different methods to measure cell viability (MTT and NR assays). What is the difference between them? They should also justify why the results are different depending the assays used. Finally, I don´t think MTT results should be presented as absorbances.

We justified the difference between methods used to measure cell count and obtained results (materials and methods and results section).

MTT results may be presented as absorbance:

Moravec RA, Niles AL, et al. Cell Viability Assays. 2013 May 1 [Updated 2016 Jul 1]. In: Sittampalam GS, Coussens NP, Brimacombe K, et al., editors. Assay Guidance Manual [Internet]. Bethesda (MD): Eli Lilly & Company and the National Center for Advancing Translational Sciences; 2004-. Figure 3: [A comparison of using the...]. Available from: https://www.ncbi.nlm.nih.gov/books/NBK144065/figure/mttassays.F3/

or

Brohawn DG, O'Brien LC, Bennett JP Jr. RNAseq Analyses Identify Tumor Necrosis Factor-Mediated Inflammation as a Major Abnormality in ALS Spinal Cord. PLoS One. 2016 Aug 3;11(8):e0160520. doi: 10.1371/journal.pone.0160520. eCollection 2016.)  or percent of control cells (Myers JN, Schäffer MW, Korolkova OY, Williams AD, Gangula PR, MʼKoma AE. Implications of the colonic deposition of free hemoglobin-α chain: a previously unknown tissue by-product in inflammatory bowel disease. Inflamm Bowel Dis. 2014 Sep;20(9):1530-47. doi: 10.1097/MIB.0000000000000144.).

In our opinion results presented as absorbances are direct and more informative results. However, if Editors or Reviewers want to change it to the % of control, we will do it. 

6.- If tables are split among two pages, please repeat the heading to make them clearer for the reader. In addition, a word is missing in foot Table 1. I think it should read as follows: Different capital letters in the same Enzyme? used temperature indicates…

The tables and foot Table 1. were improved.

7.- Line 275: please correct. Globulin 11S after 65ºC treatment (IC50= 0.44 mg/ml) and not 0.34…

It was corrected. It should be 0.44 mg/ml according to the data in Table 2.

8.- Lines 282 and 283: Please check … The lowest IC50 value was determined for the albumin hydrolyzate (0.01 mg/mL). I don´t see where this value come from.

It was corrected. The lowest IC50 value was determined for the prolamin hydrolyzate (0.06 mg/ml).

9.- Results concerning cell cycle: please add the statistics along with the graphs.

We added required information.

10.- It is quite difficult to see that prolamin control, globulin 11S 65ºC and prolamin 100ºC are the samples with the highest inhibitory activity… Please, justify better and highlight the results if needed to make it clearer for the readers.

The sentences were added:

11.- Tables: I strongly suggest to use always Capital letters first and lower letters afterwards (in table 4 is just the opposite). In addition, I suggest to reduce the superscript letters (from a to f) in table 4.

Capital letters are first.

12.- Discussion section:

420-423 and 484-487 are repetitive.

Avoid general conclusions or statements that have not been confirmed such as line 489: “… the temperature-treated millet could be used in inhibition of metabolic syndrome…”

It was changed.

Lines 506-512: this is not discussion but repetition of the results

The sentences were deleted.

13.- The in vitro results concerning PAC and PAV and enzymes are clear and summarized several times along the manuscript. However, the effects on HECa10 are not clear enough and should be summarized and highlighted.

 14.- Please check references. For example, check references 4, 8, 10, complete reference 25 (Food Chem 2013. 141 (4): 3774-80

The references were checked.

15.- Line 424: correct “demonstrating”

It was corrected.

16.- Please, write in vitro in italics along the manuscript

It was corrected.

17.- Line 450: PAV does not need to be explained as it has already been explained before

It was corrected.

18.- Line 455: please correct indication for indicating

It was corrected.

19.- Delete peptide in line 515

It was deleted.

Reviewer 2 Report

The paper “The effect of temperature on the metabolic syndrome inhibitory activity and the influence of peptides derived from millet proteins on endothelial cell HECa10” is an interesting study that reports on new results regarding the effects of millet protein hydrolyzates and peptide fractions having inhibitors of ACE, α-amylase, and α-glucosidase activity. The authors have characterized the inhibitory effect and identified the sequence of these inhibitory peptides. Based on their analysis, which also includes its effect on HECa10 cells (mouse endothelial cell), this study concludes that “heat pretreatment is a good method for preparation of millet seeds to humans suffering from metabolic syndrome.” These results open new horizons for the use of millet consumption in humans. In general, the paper is well written, clearly formulated and easy to read, the Figures illustrate the text enough and help to follow the results. A short description on not being able to use a human endothelial cells will be helpful. The author should provide appropriate references especially in the method section or indicated if it is something they’ve developed in the lab.

Author Response

The paper “The effect of temperature on the metabolic syndrome inhibitory activity and the influence of peptides derived from millet proteins on endothelial cell HECa10” is an interesting study that reports on new results regarding the effects of millet protein hydrolyzates and peptide fractions having inhibitors of ACE, α-amylase, and α-glucosidase activity. The authors have characterized the inhibitory effect and identified the sequence of these inhibitory peptides. Based on their analysis, which also includes its effect on HECa10 cells (mouse endothelial cell), this study concludes that “heat pretreatment is a good method for preparation of millet seeds to humans suffering from metabolic syndrome.” These results open new horizons for the use of millet consumption in humans. In general, the paper is well written, clearly formulated and easy to read, the Figures illustrate the text enough and help to follow the results. A short description on not being able to use a human endothelial cells will be helpful. The author should provide appropriate references especially in the method section or indicated if it is something they’ve developed in the lab.

Thank you and for helpful comments. We used endothelial cells HECa10 line as a model for in vitro studies. Choice of the cell line resulted from few reasons. Firstly, HECa10 cell line in isolated from normal mice and its exhibited endothelial functionality i.e. production of angiotensin converting enzyme and of factor VIII-related antigen, upon stimulation, they express E-selectin which binds oligosaccharides containing the Lewisx determinant (Fuc alpha 3[Gal beta 4 GlcNAc beta 3Gal beta) and the MECA 79 addressin. Secondly we had a big experience with work of the HECa10 cell line what is associated with appropriate good laboratory practice procedures for cultivating and studies on the cell line At the end we want to validate our research in mice model to find if the in vitro results will be also confirmed in vivo study. Therefore we choose mice cell line. We are aware, that there exists also humans endothelial cell lines. We going to perform research on those cell lines before human clinical studies.

We look forward to hearing a positive reply.

Sincerely yours,

Anna Jakubczyk

Round  2

Reviewer 1 Report

Once revised the new version of the manuscript from Karás et al., I feel that the new version has been significantly improved. The authors have made an effort in including new comments and explanations through the text. In fact, the introduction section has now been shortened and it is more focused on the aim of the study and the discussion has been enlarged including explanations regarding endothelial cells. The title is now more appropriate and consistent with the results described. However, not additional experiments have been included (no more enzymes analyzed or more studies concerning insulin sensitivity). Other concerns are the following:

As I said in my previous review, the conclusions reported “it seems that heat pretreatment is a good method for preparation of millet seeds to humans suffering from metabolic syndromeare not supported by the results obtained and cannot be arisen without a well design in vivo study (or even a clinical trial with humans) as these in vitro actions could not mimic what happen in the organism. Thus, this sentence should be deleted or improved.

I still believe MTT values should be expressed as percentage of control or related to total cell number. Please, change them.

In table 4 there are too many superscript letters, please reduce if possible.

Author Response

Dear Reviewer,

Thank you for your useful comments and suggestions We have modified the manuscript accordingly, and detailed corrections are listed below point by point.

Once revised the new version of the manuscript from Karás et al., I feel that the new version has been significantly improved. The authors have made an effort in including new comments and explanations through the text. In fact, the introduction section has now been shortened and it is more focused on the aim of the study and the discussion has been enlarged including explanations regarding endothelial cells. The title is now more appropriate and consistent with the results described. However, not additional experiments have been included (no more enzymes analyzed or more studies concerning insulin sensitivity). Other concerns are the following:

As I said in my previous review, the conclusions reported “it seems that heat pretreatment is a good method for preparation of millet seeds to humans suffering from metabolic syndromeare not supported by the results obtained and cannot be arisen without a well design in vivo study (or even a clinical trial with humans) as these in vitro actions could not mimic what happen in the organism. Thus, this sentence should be deleted or improved.

 The sentence was deleted.

I still believe MTT values should be expressed as percentage of control or related to total cell number. Please, change them.

 The figures were improved

In table 4 there are too many superscript letters, please reduce if possible.

It was not corrected in previous version because it is not possible reduce superscript letters in table 4.